# A realistic approach to generate masked faces applied on two novel masked face recognition data sets

**Tudor Mare**
SecurifAI
tudor.mare@securifai.ro

**Georgian Duta**
SecurifAI
georgian.duta@securifai.ro

**Mariana-Iuliana Georgescu**
SecurifAI
University of Bucharest
georgescu_lily@yahoo.com

**Adrian Sandru**
SecurifAI
adrian.sandru@securifai.ro

**Bogdan Alexe**
SecurifAI
University of Bucharest
bogdan.alexe@fmi.unibuc.ro

**Marius Popescu**
SecurifAI
University of Bucharest
popescunmarius@gmail.com

**Radu Tudor Ionescu**
SecurifAI
University of Bucharest
raducu.ionescu@gmail.com

## Abstract

The COVID-19 pandemic raises the problem of adapting face recognition systems to the new reality, where people may wear surgical masks to cover their noses and mouths. Traditional data sets (e.g., CelebA, CASIA-WebFace) used for training these systems were released before the pandemic, so they now seem unsuited due to the lack of examples of people wearing masks. We propose a method for enhancing data sets containing faces without masks by creating synthetic masks and overlaying them on faces in the original images. Our method relies on SparkAR Studio, a developer program made by Facebook that is used to create Instagram face filters. In our approach, we use 9 masks of different colors, shapes and fabrics. We employ our method to generate a number of 445,446 (90%) samples of masks for the CASIA-WebFace data set and 196,254 (96.8%) masks for the CelebA data set, releasing the mask images at https://github.com/securifai/masked_faces. We show that our method produces significantly more realistic training examples of masks overlaid on faces by asking volunteers to qualitatively compare it to other methods or data sets designed for the same task. We also demonstrate the usefulness of our method by evaluating state-of-the-art face recognition systems (FaceNet, VGG-face, ArcFace) trained on our enhanced data sets and showing that they outperform equivalent systems trained on original data sets (containing faces without masks) or competing data sets (containing masks generated by related methods), when the test benchmarks contain masked faces.

# 1 Introduction

State-of-the-art systems [4, 10, 12, 14] for face recognition, verification or clustering have achieved impressive results in recent years. These systems are trained on data sets whose dimensions have rapidly increased in the last years. Indeed, the number of images of faces of different identities has grown from a few thousands in 2007 (e.g., LFW [8] contains $\sim 13,000$ images) to hundreds of thousands in 2015 (e.g., CASIA-WebFace [16] and CelebA [11] contain $\sim 500,000$ images and $\sim 200,000$ images, respectively) or even millions of images in 2021 (e.g., WebFace260M [19] and WebFace42M [19] have $\sim 260$ million and $\sim 42$ million images, respectively). All these data sets have been released in the past 15 years, with much effort being put into collecting and cleaning them. Remarkably, some recognition models perform so well that they can even spot errors in the annotations (see [14]). However, these models are trained based on the well-known machine learning assumption that training and testing examples follow the same distribution. With the COVID-19 pandemic hitting the entire world in early 2020, we have witnessed a sharp change in the way people across the globe live. People started to wear masks to cover their noses and mouths, keeping them safe from getting infected with the SARS-CoV-2 virus. Hence, the new safety measures generated a change in the distribution of face appearances, which is no longer the same as the one before the pandemic. Thus, the practical use in real scenarios of face recognition systems trained on data lacking masked faces is currently questioned. Intuitively, the discriminative power of such recognition systems, which might reside in analyzing some primary features localized around the mouth, nose or cheeks, could significantly degrade as these regions are now masked. Face masks vary in shape, color, texture or fabrics, and convey almost no information about the covered face.

The success of modern algorithms for face recognition relies heavily on training complex deep learning models [4, 10, 14] on large-scale data sets [10, 11, 16]. Unfortunately, current data sets [2, 3, 15, 18] with masked faces are a few orders of magnitude smaller than the ones containing regular faces, without masks. To the best of our knowledge, the largest data sets [3, 15] containing masked faces contain less than $10,000$ examples.

We address the lack of data sets containing images with masked faces by proposing a fully automatic method to enhance data sets containing faces without masks. Our method relies on SparkAR Studio [1], a developer program made by Facebook that is typically used to create Instagram face filters. We use SparkAR Studio to create synthetic masks and overlay them on faces in the original images. In our approach, we consider 9 different types of masks of various colors, shapes and textures (see Figure 1). Our approach can be run on any data set containing images of faces without masks. In particular, we have applied our method for generating images with synthesized masks for the CelebA [11] and CASIA-WebFace [16] data sets. We release the images with mask overlays at `https://github.com/securifai/masked_faces` along with a script that superimposes the masks over the original CelebA and CASIA-WebFace images.

We conduct a human evaluation study for which we designed a protocol to assess how realistic the generated masks look, comparing the output of different methods that synthetically generate masks and overlay them on faces. The results show that our method produces significantly more realistic images with synthesized masked faces when compared to competing methods [2, 7, 15]. We present an in-depth analysis in Section 4.

Our enhanced data sets (containing masked faces) can be used in training state-of-the-art face recognition systems in order to capture the new distribution in appearance of masked faces. We show that face recognition systems trained on the enhanced data sets attain superior performance in the matter of analyzing images with masked faces. Moreover, we present empirical evidence indicating that, when it comes to recognizing faces covered by real masks, our synthetically generated masks provide superior results compared with a baseline based on blacking out pixels, as well as the most realistic mask generation competitor [2].

To summarize, the main contributions of our paper are the following:

- We propose an automatic method for synthesizing masks, applying a series of tools and algorithms (SparkAR Studio, Dlib-ml, ColorMatcher, custom verification scripts) to ensure the realism of the generated masks.

- We release two enhanced data sets containing a total of $\sim 640,000$ examples of masked faces.

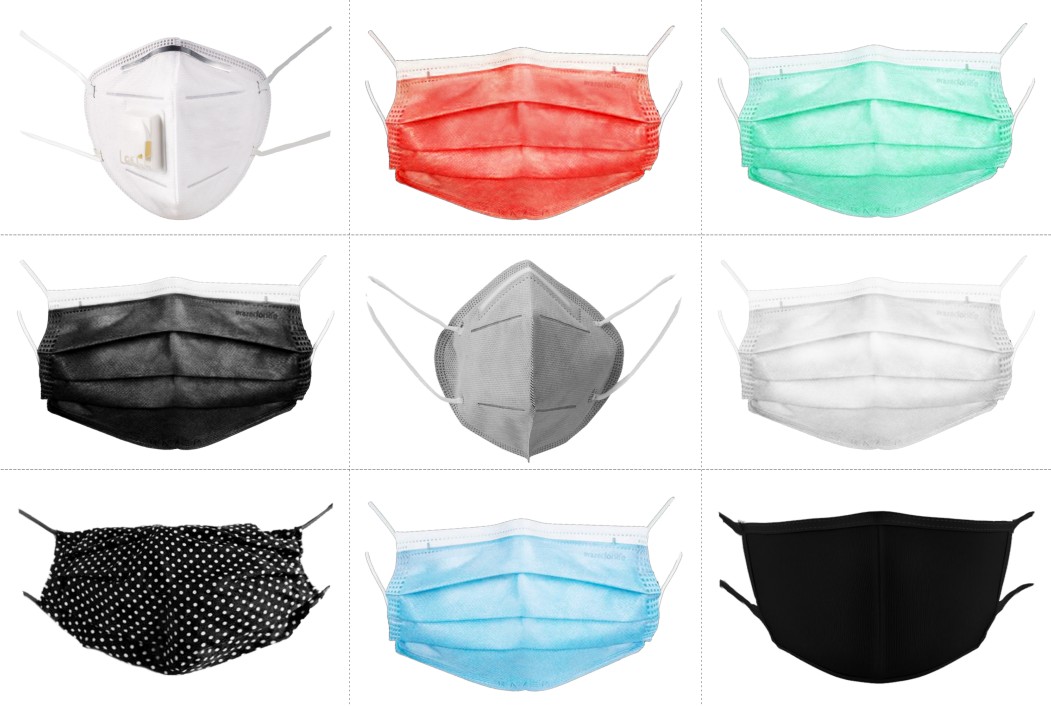

Figure 1: **Mask database.** *Our mask database contains 9 masks that vary in color, shape and texture. Best viewed in color.*

- We qualitatively compare the output of our method with the output of several competing methods in terms of how realistic the generated masked faces look.

- We show that state-of-the-art face recognition systems can perform better in recognizing people with masked faces when trained on our enhanced data sets.

## 2 Related work

The masked face recognition task is quite new in literature as it emerged as a natural problem to be solved after the COVID-19 pandemic broke out. Publicly available data sets containing real masked faces are rare [2, 15, 18] and their sizes are usually by a few orders of magnitude smaller than the large-scale data sets with regular faces. This is due to the complex pipeline of face pre-processing, cleaning the data set, and also carefully selecting images containing real masked faces such that the distribution of collected examples reflects the diversity in race, ethnicity, gender, age, pose, mask types, and so on.

The work of Wang et al. [15] was one of the earliest to propose a data set with real masked faces. Their data set, entitled the Real-World Masked Face Recognition Dataset (RMFRD), includes $5,000$ images of 525 subjects wearing masks and $90,000$ images of the same 525 subjects without masks. Anwar et al. [2] proposed the MFR2 data set, which contains a total of 269 images with 53 identities of celebrities and politicians collected from the Internet. Each identity has 5 images on average, with both masked and unmasked faces. The authors of [5] employed several data sets for testing, among them introducing the PKU-Masked-Face data set. This data set contains $10,301$ face images of $1,018$ different identities. For each identity, there are 5 images with regular faces and 5 images with masked faces with various orientations, lighting conditions and mask types. The Masked Face Recognition Challenge and Workshop organized in conjunction with ICCV 2021 proposes several benchmarks [3, 18] for deep face recognition methods under the existence of facial masks. In the InsightFace track [3], the organizers provided a test set with $6,964$ real masked facial images and $13,928$ non-masked facial images of $6,964$ identities. However, in order to avoid data privacy problems, the test set is not publicly available yet. In the WebFace track [18], the organizers provided

a test benchmark of a total of 60,926 images with masked and unmasked faces. However, there are only 3,211 images containing masked faces of 862 identities.

With algorithms relying heavily on large-scale data sets for training and with the lack of publicly available data sets with real masked faces, researchers have tried to find alternative ways for solving this issue. A simple alternative is to synthetically generate masks and overlay them on faces. This approach holds the promise of producing large-scale synthesized masked face data sets based on automatic algorithms. Wang et al. [15] proposed a simple algorithm based on the Dlib-ml library [9] for overlaying synthesized masks on faces. They generated the Simulated Masked Face Recognition Dataset (SMFRD) containing 500,000 faces of 10,000 subjects, by applying their algorithm on the LFW [8] and CASIA-WebFace [16] data sets. Anwar et al. [2] proposed the open-source tool entitled MaskTheFace to mask faces in images. MaskTheFace relies on the face landmark detector from the same library, Dlib-ml, in order to estimate the face tilt and identify six key facial features needed for applying the mask. The face tilt is used to select a template mask with a pre-defined orientation, from a database of 7 mask types, 3 orientations and 27 textures. The six key features of the face are needed at aligning the selected mask such that it fits on the face. The work of Huang et al. [7] introduced Webface-OCC, a simulated occluded face recognition data set with 804,704 face images of 10,575 subjects. This data set is built on the CASIA-WebFace data set, combining faces with simulated occlusions with their original unmasked counterparts. The occluded faces are obtained by covering the original faces with a range of occluding objects (e.g., glasses, masks) that vary in texture and color. Ding et al. [5] applied a similar data augmentation method for training data, automatically generating synthetic masked face images on LFW. They used the Delaunay triangulation algorithm to overlay different masks on the faces based on the automatically detected facial landmarks.

In Section 4, we compare our approach with competing methods [2, 7, 15] in terms of how realistic the synthetically generated face masks are, showing that our approach generates significantly more realistic images of masked faces. This is the main differentiating factor from our competitors.

## 3    Synthetic generation of masked face data sets

Our method for synthetic generation of masks relies on SparkAR Studio [1]. SparkAR Studio is a developer program made by Facebook that enables users to create augmented reality effects. Its practical use resides in creating Instagram face filters. In particular, SparkAR can be used to detect a 3D face in an input image and then apply highly realistic shadings and shadow effects on the detected face. Based on these features, we consider SparkAR to be an useful tool in creating realistic synthesized masks and overlaying them on real faces. We employ it to enhance the CASIA-WebFace [16] and CelebA [11] data sets with masked faces, as detailed below.

### 3.1    Synthetic mask generation

For a given input image containing a face, our goal is to create a synthesized mask that looks as realistic as possible when overlaid on the face, as illustrated in Figure 2(d). Our mask database consists of 9 masks which vary in color, shape and texture (see Figure 1).

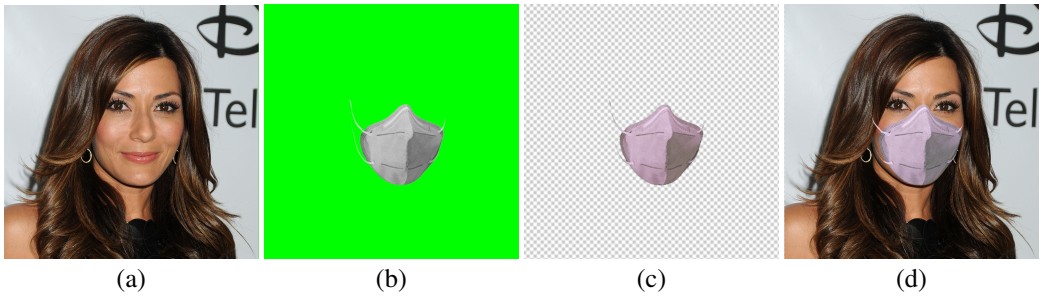

| (a) | (b) | (c) | (d) |

Figure 2: **Overview of our approach.** *For the original image (a), we select a random mask and employ SparkAR to transform it (b) based on the 3D face model. We adjust the color of the mask (c) and thus obtain the output of our method. Further, we can overlay the mask (c) over the original image to obtain the masked face image (d). Best viewed in color.*

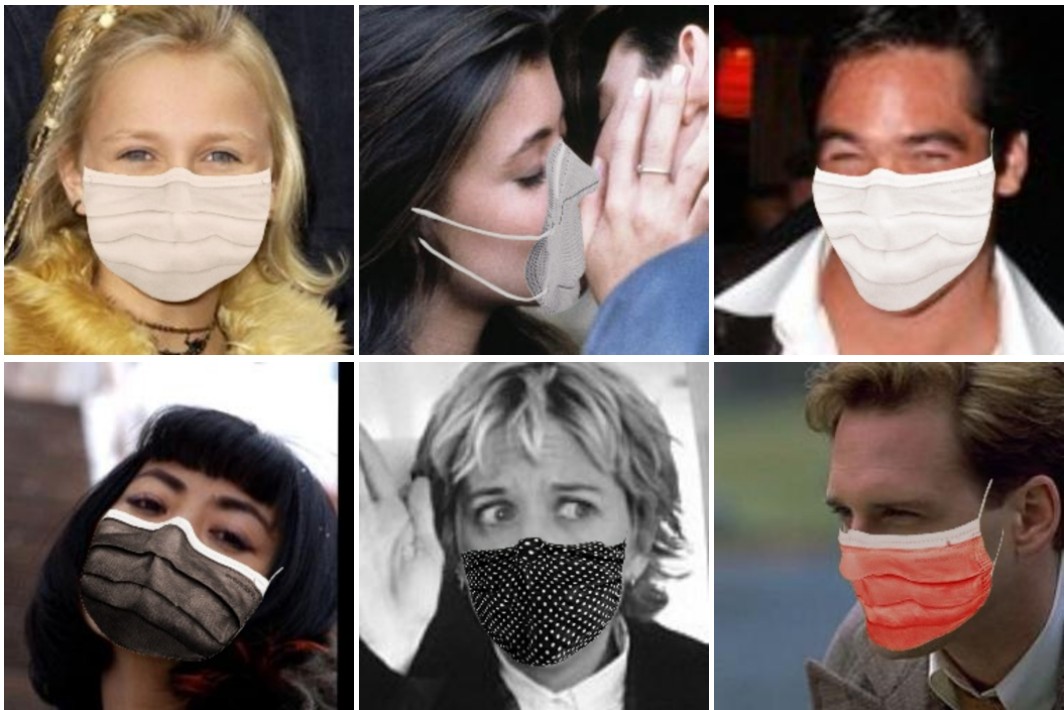

Figure 3: **Examples showcasing key features of our method.** *Our method relies on SparkAR to correctly predict face positions for various poses of the subjects, even when they are looking in a direction perpendicular to the camera. In such cases, the corresponding parts of the mask are automatically occluded. SparkAR can also recognize facial expressions and modify the 3D mask accordingly, e.g. stretching the mask when appropriate. Color matching the original image helps to blend in the mask over the context image, thus creating a more realistic sample. Best viewed in color.*

We employ the SparkAR software to detect the 3D face in the input image and randomly select a mask out of the existing 9 masks in the database to be mapped over the 3D face. Next, we use SparkAR to transform the selected mask such that it fits well on the 3D face model and then save the processed mask using a green screen background (see Figure 2(b)). This step is necessary to avoid publicly sharing any data from the original CelebA and CASIA-WebFace data sets, which are protected by license agreements that do not permit data redistribution. In order to overlay the mask on the original face, we subtract the green background and compute color matching between the segmented mask and the face. For this step, we use the Color Matcher plugin provided by After Effects. The transformed mask with transparent background and adjusted colors is the output of our method. An output example can be visualized in Figure 2(c). By overlaying the mask on the original image, we obtain the masked face image, as shown in Figure 2(d). To make sure that the masks generated with SparkAR are positioned well, we employ a verification step based on a method that detects facial landmarks, which is implement in Dlib-ml [9]. Having the facial landmarks, we derive a polygon that should roughly coincide with the region covered by the mask. Then, we compute the Intersection over Union (IoU) between this polygon and the created mask. If the computed IoU has a value lower than a fixed threshold ($0.3$ for CelebA and $0.5$ for CASIA-WebFace), we reprocess the image by employing the Dlib-ml library to apply the mask over the input image.

Even though SparkAR is very powerful, there are images on which SparkAR cannot recognize the 3D face. In this case, we also fallback to the Dlib-ml library to obtain the synthesized mask.

**CelebA+masks.** For CelebA, we consider the keypoints provided with the data set to verify if each mask produced by SparkAR (or Dlib-ml) is correctly placed over the corresponding face. Following the described protocol, we generate the *CelebA+masks* data set with $196,254$ synthesized masks, which represents $96.8\%$ of the total number of images in the CelebA data set. About $5.4\%$ of these masks are obtained using the Dlib-ml library. We underline that there are some cases when both

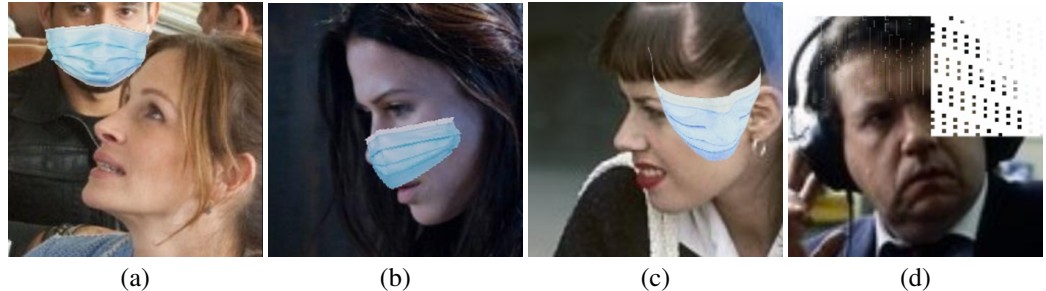

|          (a)          |          (b)          |          (c)          |          (d)          |

Figure 4: **Failure cases.** *In less than 3% of the images, the data sets might exhibit the following errors: (a) mask applied on wrong person, (b) mask covering the face only partially, (c) mask not covering the face, (d) compression artifacts. Best viewed in color.*

SparkAR and Dlib-ml fail to produce a mask and this accounts for the remaining 3.2% of all images in the CelebA data set.

**CASIA-WebFace+masks.** We use Dlib-ml to generate the face keypoints for the verification step and proceed in the same way (as for CelebA) to remove poor examples of generated masks. We employ our method to generate the *CASIA-WebFace+masks* data set containing 445, 446 synthesized masks, which represents about 90% of the total number of images in the CASIA-WebFace data set. Roughly 11.5% of the masks are obtained using the Dlib-ml library, which is almost double the percentage of images generated with Dlib-ml for the CelebA data set. We believe that CASIA-WebFace is much more diverse context-wise, presenting a higher difficulty for SparkAR in detecting the 3D face in the input image.

## 3.2 Key features

Our approach presents a few key features that enable synthetic generation of realistic masks. SparkAR proves to be a powerful tool, as it correctly recognizes the 3D position and rotation of faces in almost 90% of the processed images. This makes our approach robust and flexible compared to other approaches [2] that are bounded to using a number of predefined directions of the head, by design. From the samples illustrated in Figure 3, we can infer that SparkAR can predict the face position even when the subject is looking in a direction perpendicular to the camera. SparkAR also recognizes facial expressions and modifies the 3D mask accordingly. Therefore, we can expect the mask to cover the subject's face while they are laughing, smiling, yawning, etc. The 3D face mask can also produce occlusions on itself. Thus, when someone turns the head at an angle, a certain part of the mask may get occluded if it is not visible from the camera's viewpoint, creating a realistic image. We place the masks over the context using a smoothed alpha channel. Through this procedure, we eliminate the white jagged edges that appear when applying the mask over the original image. Color matching the images helps to blend in the mask over the context image, thus creating a more realistic sample. Because our mask images have 4 channels (RGBA), we can apply any changes to the color channels and we can use the alpha channel as a segmentation mask when training machine learning models.

## 3.3 Limitations

Even though we employ a verification step for our generated masks using Dlib-ml, failure mask generation cases can still be present in our generated data sets, as shown in Figure 4. The poor samples can be caused by multiple factors: (i) SparkAR might detect another face than the one of interest (see Figure 4(a)); (ii) SparkAR gets an inaccurate 3D face model and the transformed mask covers the face of interest only partially (see Figure 4(b)) or not at all (see Figure 4(c)); (iii) while the SparkAR software generates masks, we employ a third-party recording software (OBS) to record the screen, which might introduce unwanted artifacts, such as the one illustrated in Figure 4(d). We manually inspected a randomly sampled subset that accounts for about 1% of the generates images from each of the two data sets. Based on the manual annotation of two of the paper co-authors, we estimate that CASIA-WebFace+masks contains about 2.83% failure cases, while CelebA+masks contains about 2.90% failure cases. We underline that this problem is also prevalent in related data sets of masked faces, although not mentioned in the introductory papers. In our case, we tried to

| Comparison | Annotator ID | Votes for ours | Votes for competitor |
|---|---|---|---|
| | Person #1 | 200 | 0 |
| | Person #2 | 196 | 4 |
| | Person #3 | 197 | 3 |
| Ours versus [15] | Person #4 | 191 | 9 |
| | Person #5 | 189 | 11 |
| | Person #6 | 191 | 9 |
| | Overall in % | 97.00% | 3.00% |
| | Person #1 | 137 | 66 |
| | Person #2 | 191 | 9 |
| | Person #3 | 190 | 10 |
| Ours versus [2] | Person #4 | 177 | 23 |
| | Person #5 | 177 | 23 |
| | Person #6 | 182 | 18 |
| | Overall in % | 87.83% | 12.17% |
| | Person #1 | 194 | 6 |
| | Person #2 | 199 | 1 |
| | Person #3 | 200 | 0 |
| Ours versus [7] | Person #4 | 197 | 3 |
| | Person #5 | 192 | 8 |
| | Person #6 | 200 | 0 |
| | Overall in % | 98.50% | 1.50% |

Table 1: Comparative results showing the number of votes for our approach versus the number of votes for each of the three state-of-the-art methods [2, 7, 15]. Each comparative study is based on 200 sample pairs and it was completed by 6 human annotators.

minimize this percentage by employing SparkAR and Dlib-ml, in a cascaded fashion. We would like to add that the faulty images illustrated in Figure 4 do not represent the actual distribution of fault types. Indeed, most faults (around $90\%$) are caused by masks not covering the entire face, this being the most acceptable type of error, in our opinion.

## 4   Quantitative and qualitative evaluation of mask generation methods

Automatically generating realistic images with synthesized masks overlaid on faces holds the promise to address the lack of large-scale data sets of masked faces.

We qualitatively compare our approach to the methods presented in [2, 7, 15] in terms of how realistic the generated images with synthesized masks overlaid on faces are. For each of the three comparisons, we select a random subset of 200 image pairs. Each image pair consists of two images with synthesized masks overlaid on faces. One image is the output of our method, while the other image is the output of one of the three methods [2, 7, 15] that we compare with. The position of the images displayed in the application interface (left-hand or right-hand side) is randomly picked every time, such that the annotator does not know which method is shown in the left-hand side or in the right-hand side.

When presented with an image pair, annotators are asked to select the image that looks more realistic. For our method, we consider a fixed set of 200 images containing masked faces for all three comparisons. The set is randomly chosen from CelebA+masks. For Anwar et al. [2], we run their open-source software, MaskTheFace, on the randomly chosen subset of 200 images. For the other two competing methods [7, 15], which do not share code to produce masked faces, we randomly select 200 images from their released data sets. More precisely, we take images from the Simulated Masked Face Recognition [15] and the Webface-OCC [7] data sets.

We collect data from 6 impartial annotators and obtain a total of $3,600$ annotations (6 annotators $\times$ 200 image pairs $\times$ 3 comparative studies). Table 1 shows quantitative results of our human evaluation study. For each comparison and annotator, we show the number of votes for each of the two methods involved in the comparison. When compared to [7, 15], the annotators picked our method in more than $97\%$ of the times (on average), which is remarkable. Among our three competitors, Anwar

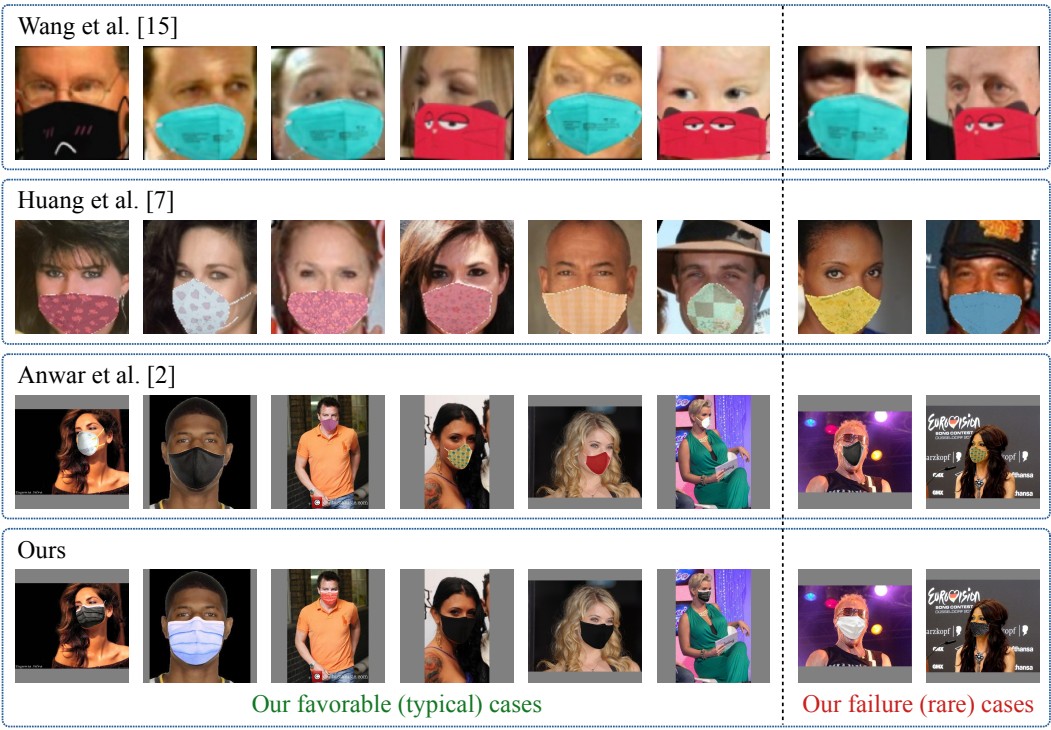

Figure 5: **Comparing the degree of realism of images with synthetically masked faces.** *We show six favorable cases and two failure cases for our comparative study. Our images (last row) are compared with the images produced by the competitors [2, 7, 15] (first three rows). Image pairs are shown column-wise. Examples shown on the first six columns represent the usual cases, when our method is superior to the other methods (see Table 1). Examples shown on the last two columns represent rare cases, when our method is inferior to one of the competitors. Best viewed in color.*

et al. [2] gained the highest number of votes. Still, our method was voted in the majority of cases (87.83%) when it is was compared to that of Anwar et al. [2]. According to the annotators, we conclude that our method produces significantly more realistic examples of faces with masks.

In Figure 5, we illustrate a set of typical cases when our method is considered superior in producing more realistic images than the competitors, along with some rare cases when our method is inferior to one of the competitors, especially Anwar et al. [2]. From the displayed examples, we observe that the method of Wang et al. [15] does a very poor job at aligning the masks to the corresponding faces, while the method of Huang et al. [7] fails to properly blend in the mask, leaving clear patterns indicating that the overlaid masks are not realistic. Confirming the results in Table 1, the masked faces shown in Figure 5 indicate that Anwar et al. [2] is the strongest competitor. Nevertheless, our method is able to more realistically align and blend in the masks. In summary, we underline that the qualitative results shown in Figure 5 are in perfect agreement with the quantitative results presented in Table 1.

## 5 Experimental evaluation of face recognition systems

We evaluate the practical usage of our method by fine-tuning three face recognition systems from the literature, namely FaceNet [14], VGG-face [12] and ArcFace [4], on masked face data sets generated by our method. We then test the face recognition models on data sets with faces covered by synthesized or real masks. We conduct our evaluation in the scenario of face verification: given a pair of two faces $(i,j)$, the task is to decide whether faces $i$ and $j$ are of the same identity or not.

**Data sets.** As training data, we consider the training splits of the CelebA and CASIA-WebFace data sets, but the images are replaced with our masked faces from CelebA+masks and CASIA-WebFace+masks. Similarly, for testing, we consider the test splits of CelebA and CASIA-WebFace,

replacing the original images with the corresponding ones from our data sets. Additionally, we report results on the MFR2 real masked face data set introduced by Anwar et al. [2].

**Evaluation protocol.** We follow the standard protocol for *unrestricted*, *labeled outside data*, as mentioned in [8], to evaluate the image pairs. The $L_2$ distance threshold between two embeddings is calculated on the nine folds of the underlying test set and used on the tenth test fold to report the accuracy, in a 10-fold cross-validation manner.

**Performance metrics.** We adopt the notation in [14] and consider, for a given test set of image pairs, $\mathcal{P}_{\text{same}}$ to be the set of all face pairs $(i,j)$ of the *same* identity and $\mathcal{P}_{\text{diff}}$ to be the set of all face pairs of *different* identities. The set of pairs correctly classified as *same* at threshold $d$ is the set of *true accepts* defined as: $\text{TA}(d) = \{(i,j) \in \mathcal{P}_{\text{same}}, D(x_i, x_j) \leq d\}$, where $x_i$ and $x_j$ are the neural embeddings of the faces $i$ and $j$ and $D$ is the squared $L_2$ distance. The set of pairs incorrectly classified as *same* at threshold $d$ represents the set of *false accepts*, which is defined as: $\text{FA}(d) = \{(i,j) \in \mathcal{P}_{\text{diff}}, D(x_i, x_j) \leq d\}$. For the threshold $d$, we select the value that provides the maximum accuracy on the validation folds, considering values in the range $[0, 4]$ with a step of $0.01$. We compute the accuracy on the test fold for the optimum threshold.

**Implementation details.** Prior to the fine-tuning and evaluation of the face recognition systems, we apply the MTCNN [17] face detector, choosing the largest bounding box whenever there are multiple face detections in an image. We apply face registration only for ArcFace. The face recognition systems are trained using the original hyperparameters reported in the introductory works [4, 12, 14]. For FaceNet [14], we select the Inception-ResNet-v1 backbone, while for ArcFace [4], we select the ResNet-50 backbone.

**Evaluation on the CelebA+masks data set.** After applying our method to enhance the data set with masks, we obtain $158,256$ training faces of $8,190$ identities and $19,494$ test faces of $1,000$ identities. We train the face recognition systems on the train split in two scenarios: on original images when no face masks are provided, and on images modified by our method with synthesized masks. We test the face recognition systems on the test split of images with synthesized masks.

**Evaluation on the CASIA-WebFace+masks data set.** After applying our method to enhance the data set with masks, we obtain $264,168$ training faces of $6,345$ identities and $93,235$ test faces of $2,115$ identities. We train the face recognition systems in an analogous manner to the CelebA+masks case. We evaluate the face recognition systems on the test split of images with generated masks.

**Evaluation on the MFR2 data set.** We apply the CelebA models on the real images from the MFR2 [2] data set. On this data set, we consider two additional baselines for each face recognition model, one that is fine-tuned on CelebA+MaskTheFace [2], and one that is fine-tuned on our mask segments with black pixels instead of textures.

**Quantitative results.** The results of the face recognition systems are presented in Table 2. While there is a clear ranking between the evaluated face recognition models, the best one being ArcFace and the least competitive one being VGG-face, we observe that fine-tuning on realistically generated faces is beneficial in each and every case. The improvements brought by our approach are always above $2\%$. We thus conclude that our approach is useful in increasing the performance of face recognition systems on masked faces.

To assess the usefulness of our mask generation approach with respect to existing masked face data sets in training state-of-the-art masked face recognition systems, we conduct an additional experiment on MFR2. Accordingly, we use the MaskTheFace open-source tool of Anwar et al. [2] (which is the second-most realistic according to Table 1) to generate synthetic masks for the CelebA data set, obtaining a set of $194,614$ masked face images. We train the three face recognition models on the resulting data set (CelebA-train+MaskTheFace) and report the accuracy on the MFR2 test set (containing real faces). The VGG-face trained on CelebA-train+MaskTheFace yields an accuracy of $88.35\%$, which is nearly $3\%$ lower than the accuracy of the VGG-face trained on our masked faces. We observe a similar pattern when training FaceNet and ArcFace on CelebA-train+MaskTheFace instead of our version of CelebA-train, the only difference being that the performance gap is around $1\%$ for both models. This clearly indicates that having more realistic face masks is useful.

We also carry out an experiment by training the face recognition models on a version of CelebA-train that uses our mask overlays, but replaces all pixels inside the mask with the black color. We test the resulting models on MFR2, observing that there are significant performance drops (around $3\%$) for

| Train set | Test set: CelebA-test+masks (synthetic) | | |
|---|---|---|---|
| | FaceNet | VGG-face | ArcFace |
| CelebA-train (no masks) | $90.96\% \pm 1.13\%$ | $84.56\% \pm 1.40\%$ | $91.78\% \pm 0.56\%$ |
| CelebA-train+masks (ours) | $93.58\% \pm 0.82\%$ | $91.51\% \pm 0.97\%$ | $95.43\% \pm 0.78\%$ |
| | Test set: CASIA-WebFace-test+masks (synthetic) | | |
| CASIA-WebFace-train (no masks) | $84.21\% \pm 1.49\%$ | $79.65\% \pm 1.81\%$ | $87.95\% \pm 1.44\%$ |
| CASIA-WebFace-train+masks (ours) | $88.06\% \pm 1.27\%$ | $86.85\% \pm 1.11\%$ | $91.47\% \pm 0.85\%$ |
| | Test set: MFR2 (real) | | |
| CelebA-train (no masks) | $92.20\% \pm 1.92\%$ | $84.29\% \pm 4.00\%$ | $91.39\% \pm 3.64\%$ |
| CelebA-train+black pixel masks | $93.04\% \pm 2.33\%$ | $88.31\% \pm 3.08\%$ | $94.22\% \pm 2.31\%$ |
| CelebA-train+MaskTheFace [2] | $95.16\% \pm 1.88\%$ | $88.35\% \pm 2.61\%$ | $94.33\% \pm 1.55\%$ |
| CelebA-train+masks (ours) | $96.22\% \pm 1.91\%$ | $91.26\% \pm 2.19\%$ | $95.16\% \pm 2.90\%$ |

Table 2: Face recognition results of FaceNet, VGG-face and ArcFace on masked faces, when the models are trained on original and masked face images. The accuracy rates are reported for both synthetically generated as well as real masked faces.

two face recognition models, namely FaceNet and VGG-face. We believe that this observation can be explained by the fact that the actual masks reveal some information about the generic shape of the lower part of the face, which is useful for face recognition systems.

**Discussion.** A relevant aspect to discuss in the context of masked face recognition is the performance level of models fine-tuned on masked faces, when the evaluation is performed on normal (unmasked) faces. Most likely, the performance on normal faces will not be maintained to the level attained by pre-trained face recognition models. However, we do not consider this as a problem, since we can employ a binary classifier to distinguish between normal and masked faces, and then use the appropriate face recognition system. Our preliminary results based on a MobileNetV2 architecture [13] show that we can achieve an accuracy of $99.44\%$ for classifying normal faces versus masked faces (synthesized or real). We believe this guarantees the same level of performance, regardless of the occlusion level (mask or no mask).

In addition to the utility for face recognition systems, we underline that our masked face data sets can be used to train models for other tasks as well, e.g. masked face detection or facial landmark detection of masked faces. This significantly broadens the applicability of our work.

# 6 Conclusion

In this paper, we proposed an automatic method for the synthetic generation of masks using SparkAR Studio for the goal of enhancing data sets with masked faces. We showed that our approach is superior in producing more realistic images than competitors [2, 7, 15]. For the face verification task, we demonstrated that face recognition systems [4, 12, 14] fine-tuned on masked faces perform better than the ones trained on regular faces, when the evaluation is conducted on masked faces. We also showed that the usefulness of masked faces for face recognition systems is proportional to the degree of realism of the synthesized masks.

In future work, we aim to propose more elaborate ways of taking advantage of our synthetically generated masked face data sets to improve face recognition models. One approach could be to consider knowledge distillation, as in [6].

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
