# OpenReview forum: "A realistic approach to generate masked faces applied on two novel masked face recognition data sets"
_NeurIPS.cc/2021/Track/Datasets_and_Benchmarks/Round2 — NeurIPS 2021 Datasets and Benchmarks Track (Round 2)_

### Official Review · Reviewer_TCbc · 2021-09-18
**Synthetic dataset of masked faces**

**Rating:** 6
**Confidence:** 3
**Correctness:** Yes
**Clarity:** Yes

**Strengths:**

This seems to be a very effective way to create synthetic data for training network to recognize faces with masks. This significantly shortens the time for developing algorithms for recognizing faces with masks.

**Weaknesses:**

The technical contribution of this paper seems to be limited as the authors simply apply an off-the-shelf algorithm to generate the synthetic data.

The authors should provide more ablation study to demonstrate how the synthetic masks affect the performance of the network on real world dataset. For example, will the network perform better if we have more variety in the synthetic masks? Can we simply mask out part of the human face by random rectangle? I think it is good that the authors provide a qualitative evaluation where they ask human which synthetic data looks more realistic. But I do not think that is the best way to evaluate whether the dataset is good enough. The authors should focus more on the performance on the real world dataset because our goal is to perform well in the real world scenarios.

**Additional Feedback:**

Not applicable

**Documentation:**

Yes

**Ethics:**

The authors simply overlay synthetic masks on existing face images so I do not think there is any ethical concern.

**Relation To Prior Work:**

Yes

**Summary And Contributions:**

This paper proposes a synthetic dataset of masked faces. They use a program which is used to create face filter by Facebook to overlay synthetic masks on real world face images. They show that their data is more realistic than existing approaches by a qualitative evaluation and the network pre-trained on their data outperforms networks pre-trained on existing synthetic masked data by evaluating them on both synthetic and real world dataset.

---

### Official Review · Reviewer_hp7D · 2021-09-21
**Adding realistic synthetic masks to existing large-scale face recognition datasets**

**Rating:** 5
**Confidence:** 4
**Correctness:** This seems satisfactory to me.

**Strengths:**

Face recognition has long been a popular topic in the vision community. The recent pervasiveness of masks worldwide makes this certainly a timely subject. The paper demonstrates that the new synthetically masked faces look more realistic to human judges than those of previous datasets. The paper is written clearly.

**Weaknesses:**

Although the synthetically masked faces look more realistic than previous datasets of synthetically masked faces, it is not clear whether training on the new dataset will yield better performance on real masked face identification than training on the existing datasets. I suspect that it may not help, because what a face recognition system primarily learns from masked faces is to ignore the masked part of the face. A slightly more realistic mask is unlikely to provide much advantage, because the it is just as easy to learn to ignore a less realistic mask. Perhaps the size of the new dataset will provide it an advantage for training, but to demonstrate that, it would need to be compared against systems trained using each of the previous three masked datasets.

The proportion of images (about 3%) on which the synthetically masked images are incorrect is quite high for a released dataset.
The large majority of those mistakes could probably be fixed automatically with clever use of various existing vision algorithms, and this should be done before releasing the dataset. For example, in Figure 4, examples (b) and (c) are probably incorrect because dlib does not do well with profile faces. There are available deep face alignment algorithms that work well with profile faces, which could be helpful to clean this up.

The heavy use of an existing commercial package (SparkAR) does not make the dataset less useful, but it does reduce the novelty of the paper.

It is not clear how the Color Matcher plugin modifies the color of the mask to better fit the color of the image. E.g., does it normalize the histogram of the mask to match that of the image? Is that the right thing to do for optimal realism?

Are the 9 mask styles enough? This could be tested, for example, by comparing test performance vs. the number of training mask styles.

Does fine-tuning on masked faces maintain the same level of performance as the pretrained model on normal (unmasked) faces? This should also be tested.

**Additional Feedback:**

See previous answers.

**Clarity:**

The paper is very clearly written.

A few minor corrections:
- Abstract lines 11-12: Change decimal points to commas in the middle of the large numbers.
- Line 95: change "no" to "not" publicly available.


**Documentation:**

This submission is missing the required pdf in the supplementary material.

The details of the fine tuning are not provided.

**Ethics:**

This seems sufficient.

**Relation To Prior Work:**

Citing of prior work seems adequate, though not exemplary.

**Summary And Contributions:**

This work provides synthetic mask overlays for existing large-scale face recognition datasets. The overlays were generated using Facebook's Spark AR Studio, as well as dlib for additional alignment. The resulting masked images are a clear subjective improvement over synthetic masked overlays in previous approaches, as shown by a quantitative human preference study.

However, there is no experiment evaluating whether these more realistic-looking images actually lead to an improvement over previous datasets on the metric that really matters: does training on the new dataset provide better face recognition performance on real masked images than training on the older (less realistic) datasets? Overall, this is a useful contribution, but I do not think it reaches the threshold of novelty and significance required for NeurIPS.

Note: After the authors' rebuttal, I raised my rating from 4 to 5. I still believe that the novelty and originality are below the bar for NeurIPS. Please see my response to the rebuttal for details.

---

### Official Review · Reviewer_ud8T · 2021-09-22
**A nice idea and an interesting dataset.**

**Rating:** 8
**Confidence:** 3
**Correctness:** The construction is sound and transpa…
**Clarity:** The paper is clear and east to follow.

**Strengths:**

The authors use a neat method to solve a current and pressing problem. Their method to generate masked faces seems to be smart and useful. The released datasets seem to be well structured and useful for future research.

**Weaknesses:**

The authors give a ~2.5% rate of faulty images - it would be useful to have some community process to improve the dataset and report broken images.

**Additional Feedback:**

I hope we don't need this dataset much longer...

**Documentation:**

All is well documented on the github page. The links to the actual dataset could be a bit more prominent on the github page.

**Ethics:**

Since the face data is already public, I see no ethical problems, except the usual problems with facial recognition technology.

**Relation To Prior Work:**

They show other approaches and compare their results to them.

**Summary And Contributions:**

The authors present a method to create masked faces out of an existing database of faces. This is of course useful in the current pandemic to evaluate how facial recognition tools react to this new data.

---

### Decision · Program_Chairs · 2021-10-09

**Decision:**

Accept

**Comment:**

The data and methods provided in the paper are of relevance for the NeurIPS data track. Based on reviewers opinions, and in particular after discussion with the authors, the paper achieves the minimum score required for publication at NeurIPS data track.